# Between Foragers and Farmers: Climate Change and Human Strategies in Northwestern Patagonia

**Adolfo F. Gil [1,2,\*], Ricardo Villalba [3], Fernando R. Franchetti [4], Clara Otaola [1,2], Cinthia C. Abbona [1], Eva A. Peralta [1] and Gustavo Neme [1]**

1   Instituto de Evolución, Ecología Histórica y Ambiente (IDEVEA-CONICET & UTN FRSR), Av. Urquizas 350, San Rafael, Mendoza 5600, Argentina; claraotaola@gmail.com (C.O.); abbonacinthia@gmail.com (C.C.A.); evaailenperalta@gmail.com (E.A.P.); gneme@mendoza-conicet.gob.ar (G.N.)
2   Departamento de Historia, Facultad de Filosofía y Letras, Universidad Nacional de Cuyo, Mendoza 5500, Argentina
3   Instituto Argentino de Nivología, Glaciología y Ciencias Ambientales (IANIGLA-CONICET), Mendoza 5500, Argentina; ricardo@mendoza-conicet.gob.ar
4   Departamento de Historia, Facultad de Historia, Facultad de Filosofía y Letras, Universidad Nacional de Cuyo, Mendoza 5500, Argentina; ferfranchetti@gmail.com
*   Correspondence: agil@mendoza-conicet.gob.ar

**Abstract:** In this paper we explore how changes in human strategies are differentially modulated by climate in a border area between hunter-gatherers and farmers. We analyze multiple proxies: radiocarbon summed probability distributions (SPDs), stable C and N isotopes, and zooarchaeological data from northwestern Patagonia. Based on these proxies, we discuss aspects of human population, subsistence, and dietary dynamics in relation to long-term climatic trends marked by variation in the Southern Annular Mode (SAM). Our results indicate that the farming frontier in northwestern Patagonia was dynamic in both time and space. We show how changes in temperature and precipitation over the last 1000 years cal BP have influenced the use of domestic plants and the hunting of highest-ranked wild animals, whereas no significant changes in human population size occurred. During the SAM positive phase between 900 and 550 years cal BP, warmer and drier summers are associated with an increase in $C_4$ resource consumption (maize). After 550 years cal BP, when the SAM changes to the negative phase, wetter and cooler summer conditions are related to a change in diet focused on wild resources, especially meat. Over the past 1000 years, there was a non-significant change in the population based on the SPD.

**Keywords:** farming border; Patagonia; Southern Annular Mode; late Holocene; human diet; zooarchaeology; stable isotopes; farming spread; low-level food production

## 1. Introduction

For most archaeologists, farming was an evolutionary, progressive, and irreversible process [1–4]. From this perspective, farming's origin and expansion led to population increases and occupation of new areas. Farming expanded in a continuous process and only stopped when the environment made farms impossible [5,6], defining a stable frontier that persisted and that would have persisted in time. In historical times, some of these borders were occupied by foragers, but this last scene is masked and omitted in the models [2,7]. Recent research demonstrated that the border area among hunter-gatherers and farming societies never was the case for a lineal non-reversible expansion [8–10]. Both subsistence strategies were reversible, and even complementary on incipient farming. Therefore, our objective is to understand the stability of agricultural borders. Under which conditions did the border area perpetuate

itself? Was the border unstable and variable over long periods of time (e.g., millennia, centuries)? This paper evaluates the degree to which climatic variations could modulate a retraction in the farming borders. We will focus on northwestern Patagonia, where researcher proposed a pre-Hispanic farming border [11,12].

In the northern sector of the region, the land consisted of enclaves of groups with an agricultural component in their subsistence, beginning ca. 2300 years BP [13–15]. The rest of northwestern Patagonia was the land of the hunter-gatherers [11]. In historic chronicles, only hunter-gatherers are mentioned within southern Mendoza [16,17]. However, mostly inferred by domestic plant macrorests, during pre-Hispanic times, farmers might have occupied terrains farther south than the farmer/hunter-gatherer border observed during historical times. If this assertion is right, there was a retraction of farming after it reached its maximum expansion. This contradiction between the historical and archaeological records cast doubt on the idea of a stable long-term border.

In northwestern Patagonia, previous research focused on the last 2000 years cal BP, proposing and discussing an intensification process that developed from the middle of the late Holocene [18]. Most of this research focused on a multi-millennial time scale, assuming few changes after the intensification process started to develop [18–21]. This lack of fine-grain research on archaeological trends after 2300 years cal BP implicitly generated a static view of this period, implying that little changes occurred in the human-environment system. To assess how stable the farming/hunter-gatherer border was in northwestern Patagonia, we compared the temporal trends in the SAM (Southern Annular Mode) with human population dynamics (based on radiocarbon summed probability distributions (SPDs)), human diet (stable isotope ratios in human bones, namely ($\delta^{13}$C and $\delta^{15}$N), and subsistence (zooarchaeological record).

### 1.1. Climate and Environment in Northwestern Patagonia during the Last Millennium

This paper focuses on southern Mendoza, in northwestern Patagonia, where we will concentrate on two desert ecosystems: Monte Desert and Patagonia Desert (Figure 1; [22–24]). Given the biogeographical similarity between the Altoandino and Patagonia deserts and the low number of archeological samples available for the former, this work includes both groups of samples under the Patagonia Desert unit. The Monte Desert occupies the lowland plain between 200 and 1400 meters above sea level (MASL). It is a temperate region, with summer-dominant rainfall (100–350 mm per year) [22,24–26]. The Patagonia Desert is a cold, dry, and wind-swept environment. In the study area, it is restricted to a piedmont fringe bordering the Andean cordillera between 1000 and 2500 MASL (Figure 1). At 1400 MASL, when the Monte Desert limits with the Patagonia Desert, annual precipitation averages 300 mm.

The Southern Annular Mode (SAM) or Antarctic Oscillation is the main mode of climate variability in the middle and high latitudes of the Southern Hemisphere (SH), associated with approximately 30% of the precipitation and temperature variability in the austral hemisphere [27,28]. Its spatial structure comprises pressure-synchronous anomalies of opposite signs between the mid (45° S) and high (60° S) latitudes of the Southern Hemisphere. When pressures are anomalously high around Antarctica, but low at mid-latitudes, the SAM is defined as in its negative phase and vice versa. These pressure changes in the SH modulates the latitudinal position and the strength of the storm tracks associated with the Westerly winds. As a result, many extratropical regions in South America show significant changes in temperature and precipitation relative to the positive or negative phases of the SAM [27,28].

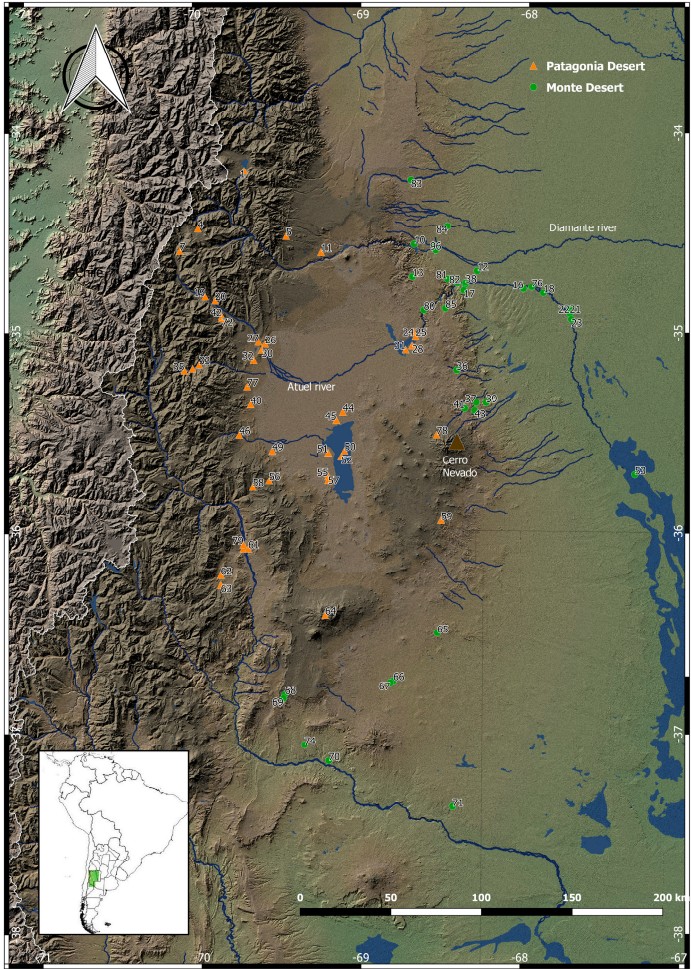

**Figure 1.** Study area with archaeological sites mentioned in the text. Archaeological sites references: 1: Laguna El Diamante; 2: Risco de Los Indios; 3: La Hedionda; 4: El Indígeno; 5: Gruta El Carrizalito; 6: Dique 25 de Mayo; 7: Los Peuquenes; 8: Los Coroneles; 9: Puesto Agua del Zapallo; 10: Los Reyunos; 11: Alero Montiel; 12: Loma del Eje; 13: El Durazno; 14: Rincón del Atuel 1; 15: Gruta del Indio; 16: MDA.114.116; 17: Cerro Negro; 18: Real del Padre 1; 19: Agua Buena; 20: Arroyo Malo 1; 21: La Olla; 22: El Bosquecillo 5; 23: El Bosquecillo 3; 24: Respolar; 25: El Nihuil; 26: El Sosneado; 27: Tierras Blancas; 28: Arbolito; 29: El Nihuil 1; 30: Las Ramada; 31: Arbolito-6; 32: Ojo de Agua; 33: Puesto Jaque 2; 34: Gendarmería Nacional 5; 35: Cueva A. Colorado; 36: Los Leones 5; 37: Agua de los Caballos; 38: Zanjón El Morado; 39: Agua de la Mula; 40: Cerro Mesa; 41: Cueva Zanjón del Buitre; 42: Arroyo Panchino; 43: Puesto Ortubia; 44: LLancanelo 22; 45: Norte de Llancanelo; 46: A Negro de Pincheira; 47: Bajada de Las Tropas; 48: Bajada de las Tropas; 49: Ea. Las Chacras; 50: Llancanelo; 51: Los Menucos; 52: Llancanelo 17; 53: Cochico; 54: Llancanelo 50; 55: Pozos de Carapacho; 56: Manqui Malal; 57: Cañada de las Vizcachas; 58: Casa de Piedra El Chequenco; 59: Ciénaga de Borbarán; 60: El Manzano; 61: Alero Puesto Carrasco; 62: A. Mechenquil; 63: El Alambrado; 64: El Payén; 65: La Peligrosa 2; 66: La Matancilla; 67: Agua del Toro; 68: Agua de Perez 8B; 69: Agua de Perez 1; 70: Médano R6; 71: Corcovo 1; 72: Cueva Palulo; 73: Campos Las Julias; 74: Carmonina; 75: Gruta Puesto Las Tinajas; 76: Los Gallegos 1; 77: Cueva Salamanca; 78: Cupertino; 79: Gruta del Manzano; 80: Cañón del Atuel; 81:Arroyo de los Jilgueros; 82: Camping Familiar Cristiano; 83: Campos Las Julias; 84: La Hedionda; 85: Puesto Aisol; 86: Laguna El Sosneado. Map base [29].

*1.2. Human Population Dynamic and Climate in Northwestern Patagonia: Background and Expectations*

Hunter-gatherers settled in northwestern Patagonia since the late Pleistocene–early Holocene [30–34]. During the late Holocene, people occupied new areas, including the most inhospitable regions of La

Payunia and the high-altitude Andes [35,36]. The intensity of land use increased in terms of both the diversity and density of archaeological assemblages during the last 2300 years cal BP [31,37–39].

Subsistence patterns changed over time, showing an increase in the range of exploited taxa [40–42]. The guanaco, a wild South American camelid, is the largest animal prey available in this area, and the best ranked in terms of energetic returns [42–45]. However, there is a decline in the abundance and significance of guanaco after ca. 2000 years cal BP [21,41]. While camelids are regular components of Monte archaeofaunas, smaller prey, such as armadillos, fish, birds, and medium/big size rodents, are also frequent [20,35,41,45]. Rheid eggs are common in both deserts, but they seem to have been more significant in the Monte Desert [46]. Archaeobotanical evidence indicates that algarrobo (*Prosopis* spp.) and chañar (*Geoffroea* spp.) were the most common taxa consumed in the Monte Desert, whereas molle (*Schinus polygamous*) was the most frequently eaten in the Patagonia Desert [47].

Based on the spatial–temporal distribution of domesticated plants, the Atuel and Diamante rivers (Figure 1) were considered as the southern boundary since ca. 2300 years BP, with "low-level food production systems" [11,15,48–50]. South of these valleys, the archaeological record includes only scattered evidence of domesticated plants in the vicinity of Cerro Nevado (*Zea mays* and *Cucurbita* spp.) [11,15,35,47]. Although microfossils of maize may have also been found in the Patagonian region of northern Neuquén [51] the only unequivocal domesticates known from the Patagonian Desert comes from macro remains specimens of bottle gourds (*Lagenaria* spp.), and are not quantitatively significant [15,52].

From an evolutionary and ecological perspective, there is a relationship between resource productivity and human population stability [2]. The origin of agriculture is explained as a human response to the unbalance between human demography and resources availability [4,50]. The stability of human populations results from interactions between climate, resources, internal population processes, life history characteristics, and social processes [2,4,53]. In terms of the population dynamic, Zahid et al. [54] argues that " … radiocarbon record shows that transitioning farming societies experienced the same rate of growth as contemporaneous foraging societies. The same rate of growth measured for populations dwelling in a range of environments and practicing a variety of subsistence strategies suggests that the global climate and/or endogenous biological factors, not adaptability to local environment or subsistence practices, regulated the long-term growth of the human population during most of the Holocene." In addition, Bettinger [2] affirms that pre-industrial agricultural populations did not grow faster than hunter-gatherer populations. This implies that the differences in adaptation, environment, and time are mild influential determinants of population growth. If climate impacted on the resources, the tradeoff between farming or hunter-gathering will change in organizational variables to maintain optimal responses in the demography-resource equation.

In the frame of the diet-breath model [55,56], we accept wild and domestic plants as low-ranked resources for northwestern Patagonia [21,43]. On the other hand, we consider the faunas, mostly guanacos, armadillos, and ostrich eggs, as high-rank resources [43]. In a scenario where population increase leads to pressure on resources, we expect a change in diet, shifting towards greater reliance on plants, including domesticated plants [21]. The diet-breadth model predicts that guanacos and armadillos, the highest ranked items, will always be in the diet. Low-rank resources, such as algarrobo and maize, should be included in the diet only when highest-rank items are scarce or in decline. Two alternative responses under domesticated use can be modeled: low and high yield density of cultigens [57]. Under a low density of domesticates, they will be a small part of the diet and a non-significant increase in human population is expected; but, in a high density cultigen context, we expect domesticates to form a significant part of the diet, with a subsequent increase in human population density.

When climatic change affects resources productivity, we can expect changes in the human resources balance and, consequently, in their diet and population size [55,56,58]. If climatic change affected farming, diminishing the productivity, different human responses can occur [4]. If domesticates are a significant part of the human diet (more than 40% or 50%), a drop in demography could be expected. Freeman et al. [53] proposed that a reversion between maize farmers and wild seed gatherers can be a response

when the domestic plant component is less than 40%. Under this scenario no significant population change can be expected. In addition, if climate increase wild resources productivity, or a human demography drop reduces pressure on resources, we would expect a focus on the highest-rank wild resources.

## 2. Materials and Methods

### 2.1. SAM Reconstruction

Paleoclimatic records for our study region are very scarce. In order to infer past climate variations in southern Mendoza, we employed a recently developed multi-proxy-based reconstruction of summer (December–January) SAM over the past 1000 years cal BP [59]. To establish the relationship between the SAM and regional climate, spatial correlation patterns were estimated across southern South America between Climatic Research Unit (CRU) temperature and precipitation, and the SAM instrumental index over the common period 1957–2016 [27]. The November–January spatial correlation patterns over South America south of 10° S latitude between temperature and precipitation were estimated using the facilities provided by the KNMI (https://climexp.knmi.nl/start.cgi). During late spring–summer (November–January), climate strongly influences the primary productivity of the desert ecosystems in southern Mendoza–northern Patagonia [60].

### 2.2. Radiocarbon Dates as Data: Summed Probability Distribution (SPD)

We monitored human population dynamics using the radiocarbon summed probability distributions (SPDs), a tool often used to reconstruct prehistoric populations [30,61,62]. This analysis is based on the "date as data" framework proposed by Rick [63]. The SPD as proxy for human population dynamics has been subject to criticism, including sampling errors, time-dependent taphonomic loss, spatial and temporal differences in site-to-population ratios, and spatial and temporal differences in sampling intensity [54,64–73]. Crema et al. [61] provide details on how the SPD limitations should be considered for obtaining consistent results.

We calculated SPD using RCarbon, an R software package for the analysis of large collections of radiocarbon dates [74]. Observed SPDs are fitted to different demographic models [74]. Random calendar years dates were generated via Monte-Carlo simulation and subsequently "back-calibrated" into $^{14}$C dates to generate null SPDs [61,75]. This approach allows the detection of statistically significant local deviations (i.e., portions of the SPD showing smaller or greater SPD than the null model) as well as the estimation of a global significance test [76]. A non-parametric extension of the hypothesis-testing approach enables the statistical comparison of two or more sets of $^{14}$C dates. The null-hypothesis is the equality of the SPDs.

This paper focuses on the last 1000 years cal BP, in which taphonomic loss can be considered as a minor factor affecting our database. Most of the dates are from samples processed in radiocarbon laboratory during the last 20 years (references in Table S1) and from research programs oriented to the study of long-term cultural process with no special temporal focus [15]. We can remark a possible bias against post-Hispanic dates as far as the Hispanic material culture have archaeological markers to date the last 500 years. If so, we can observe fewer dates for the last 500 years. But it is not the case in northwest Patagonia, where the Hispanic material culture is weakly recorded after 100 or 150 years BP [77]. We built the SPD using different spatial scales: northwestern Patagonia, and secondly divided this area into the Monte Desert and Patagonia Desert. We compared the shape of the SPDs of the two deserts with the permutation test described above, using the same calibration procedure and with 10,000 iterations. To avoid the edge effect, we used a radiocarbon database for the last 1400 years cal BP (n = 128; archaeological sites n = 78; Table S1). The radiocarbon dates come from Monte Desert (n = 60; archaeological sites n = 33) and Patagonia Desert (n = 68; archaeological sites n = 45). We included the $^{14}$C results obtained from charcoal (n = 78), plants/maize remains (n = 7), bone collagen (n = 52), leather (n = 1), and material without information. The archaeological sites were open air sites (n = 53), caves, rockshelters (n = 21), and sites without a type information (n = 4). Most of the archaeological contexts

are human burials loci (n = 34), some are domestic multi-activity (n = 37), others are specific activities sites (n = 5), and few have no information about their context (n = 2). In order to reduce the effect of bias we binned the $^{14}$C dates based on the clustering of the mean $^{14}$C years BP by archaeological site, using a threshold of 50 years.

This approach is robust to inter-regional differences in sample size, as the comparison is based on the "shape" of the SPDs and not on differences in their absolute magnitudes [74]. We first assessed whether the SPD for the northwestern Patagonia area showed statistically relevant fluctuations when compared against the uniform and the exponential null models, following the procedure described in Timpson et al. [68], using 10,000 Monte-Carlo simulations, and calibrated with the SHCal13 curve. We used "uniform" in order to detect any possible change of the SPD as significant. On the other hand, we use "exponential" assuming it helps to consider taphonomic loss [66]. To detect differences in the population dynamics between the Monte and Patagonia deserts, we ran the permTest function in Rcarbon [74,76,78].

### 2.3. Bone Collagen Stable Isotope Carbon and Nitrogen

To reconstruct the human diet, we employed stable isotopes from bone collagen and bone structural carbonate [79]. Our data comes from previously published studies of this region [31,80,81] and new samples provided in Table S2. Extraction of bone collagen and structural carbonate was performed in the Museo de Historia Natural de San Rafael [80,82]. The analyses were carried out at the Stable Isotope Ratio Facility for Environmental Research (SIRFER) at the University of Utah, Stable Isotope Facility Wyoming University and Laboratorio de Isótopos Estables para Ciencias Ambientales (LIECA).

Collagen $\delta^{13}$C and $\delta^{15}$N were determined by flash combustion to produce $CO_2$ and $N_2$. In SIRFER the resulting gases were analyzed using a Finnigan Delta Plus isotope ratio mass spectrometer (Finnigan, Bremen, Germany) coupled with a Carlo Erba Model 1110 elemental analyzer (Carlo Erba, Milan, Italy) through a CONFLO III open split interface (Finnigan, Bremen, Germany). Stable carbon and nitrogen isotopic compositions were calibrated relative to the VPDB and AIR scales using USGS40 and USGS41. Both stable isotope measurements and the sample weight percent carbon and nitrogen were obtained from a single sample combustion. Analytic precision is 0.1‰ for carbon isotope ratios (VPDB) and 0.2‰ for nitrogen (AIR). In LIECA, we followed a similar procedure using a Thermo Scientific DELTA V Advantage continuous flow isotope ratio mass spectrometer coupled via ConFlo IV to an Elementar Analyzer Flash 2000. Stable carbon and nitrogen isotopic compositions were calibrated relative to the V-PDB and AIR scales using USGS-40 and USGS-41a. Measurement uncertainty was monitored using in-house collagen standards with well-characterized isotopic compositions: Caffeine LIECA 17 ($\delta^{13}$C = −33.02‰, $\delta^{15}$N = −2.02‰), SRM-14 polar beer bone collagen ($\delta^{13}$C = −13.66‰, $\delta^{15}$N = +21.52‰), and bone collagen LIECA-17 ($\delta^{13}$C = −18.16‰, $\delta^{15}$N = +11.07‰). Precision (u(Rw)) was determined as ±0.17‰ for both $\delta^{13}$C and $\delta^{15}$N on the basis of repeated measurements of calibration standards, check standards, and sample replicates. C:N ratios are in the range of 3.1 to 3.6, indicating generally good preservation [83]. The bone carbonate was sent to the University of Wyoming Stable Isotope Facility (Gas Bench online with a Finnigan Delta Plus XP) for analysis. The ratios of stable isotopes for carbon and oxygen were determined there and reported in values related to the V-PDB standard (analytical precision 0.3‰).

As a consequence of the fractionation between plant tissues and the consumer, bone collagen was 5‰ more enriched than the dietary sources [84]. Bone collagen from individuals with a diet mostly composed of $C_3$ plants exhibit $\delta^{13}$C values between −24‰ and −19‰; individuals with a diet mostly focused on $C_4$ plants have $\delta^{13}$C values between −8.5‰ and −6.5‰ [85]. Carbon isotope signatures on bone collagen reflect mostly the protein component of the diet; they are preferentially routed from dietary sources and incorporated into body tissues [86,87]. Fernandes et al. [88] indicate that 75% of the carbon in bone collagen is derived from dietary protein; the other 25% is derived from carbohydrates and fats. Using a 9.5‰ offset diet–bone carbonate, we expect that carbonate $\delta^{13}$C signals for 100% $C_3$ and 100% $C_4$ consumers would be around −20 to −14‰ and −4‰ to −2‰, respectively.



Nitrogen isotope ratios reflect diets mediated by the metabolic pathways by which nitrogen is fixed in various tissues [86]. The diet is the primary determinant of isotope values. Most animals exhibit an approximately 3‰ shift in $\delta^{15}$N values relative to their diet by the "trophic enrichment factor" (TEF) [89,90]. Domestic plant nitrogen values vary with soil characteristics and nitrogen cycling, as do the nitrogen values of many other wild plants.

This paper adopts the following fractionation values for analysis (resources–consumer bone collagen): $\delta^{13}$C = 1.0‰ and $\delta^{15}$N = 4.6‰ for animals [91–93]; $\delta^{13}$C = 3.1‰ and $\delta^{15}$N = 4‰ for plants [91,92]. For $\delta^{13}C_{carb}$ the offsets used were $\delta^{13}$C = 8.0‰ and $\delta^{15}$N = 4.0‰ for animals [91,92] and $\delta^{13}$C = 11.1 and $\delta^{15}$N = 4.6‰ for plants. Modern samples were corrected for the Suess effect by adding 1.5‰ to their $\delta^{13}$C values [94,95].

The human bone stable isotope values database includes 43 individuals (Table S2), and only adults and juveniles are included. All samples' C:N ratios were between the accepted threshold [83]. Radiocarbon dates were calibrated using the OxCal/SHCal13 curve and reported as median ages plus or minus a 68.7‰ probability [96]. Few bone samples have chronology inferred from an associated radiocarbon date. In order to improve the explanation of human bone stable isotope values ($\delta^{13}$C and $\delta^{15}$N) in terms of the diet, we employed regional stable isotopes data on resources [25,80,97]. Using these resources as a reference framework, we explore in a more solid way the meaning of variation in $\delta^{13}$C and $\delta^{15}$N.

In addition, we explored the distribution of stable isotopes in human bone to define a spatial pattern and to discuss variations in our expectations. Q-GIS was used to generate an "isoscape" [98,99] base on $\delta^{13}C_{carb}$ and $\delta^{15}$N from human bone using the temporal frame defined by SAM. We created a raster surface model using the inverse distance weighted (IDW) interpolator model [100,101]. Interpolation was performed with a power parameter (p) equal to one, between 12 and 20 neighbors within a circular window. The maximum and minimum values on the interpolated surface occurs only on those sample dots with available data. For this paper, we used the Las Palmas 2.18 Q-GIS Geographic Information System, version 2.18.

## 2.4. Artiodactyl Index

We investigated variations in faunal consumption by measuring the variability in consumption of the best-ranked prey, the guanaco. To test if resource depression occurred in the last millennium, we use the artiodactyl index (AI), which measures the relative importance of artiodactyls in prehistoric human diets [102–105]. For each zooarchaeological assemblage, we analyzed the taxonomic structure and composition [106]. Each bone specimen was classified according to a taxonomic category and the Number of Identified Specimens (NISP) for each taxon was tallied. We estimated a set of different indices of taxonomic structure and similarity for each assemblage, such as taxonomic richness, diversity index, and evenness [45,107]. The AI considers the amount of specimens identified as guanaco, and estimates its abundance over the rest of the animal preys. It offers values between 0 and 1. An index equal to 1 indicates the dominance of the guanaco in the assemblage. Values close to 0 indicate little presence of this taxon, and as a consequence a broader diet including other taxa. This index is calculated by the formula:

AI = ΣNISP Artiodactyl/(Σ NISP Artiodactyl + NISP all other taxa)

We defined temporal analytical units for each zooarchaeological assemblage following a chrono-stratigraphic criterion. The oldest $^{14}$C date defined the lower limit, and the youngest $^{14}$C date defined the upper limit of each zooarchaeological assemblage. We considered the whole assemblage as a single temporary unit when the archaeological sites have a single date. In cases with intrusive features in their stratigraphy (i.e., Rincón del Atuel-1) [108], we considered the dated features as an independent assemblage. For sites that have more than one $^{14}$C date, which statistically overlapped (i.e., Risco de los Indios), we obtained an average value and we assigned to the assemblage a single date (Table S3)

The statistic tests were performed using Past 4.01 [109]. Temporal trends in the archaeological proxies were analyzed by plotting smooth Loess curves using the GGplot2/R [109,110].

## 3. Results

### 3.1. SAM Reconstruction for the Last 1000 Years

Based on instrumental records, the SAM positive phase associates with warmer November–January temperatures over Patagonia. These anomalies are stronger on the Atlantic coast but expands to the southern Mendoza lowlands (Patagonian and Monte deserts; Figure 2A). On the other hand, the spatial patterns of correlation between SAM and precipitation from November to January show that the SAM positive phase relates to negative deviations in precipitation over the Pacific coast at 35°–38° S, reaching southern Mendoza with less intensity (Figure 2B).

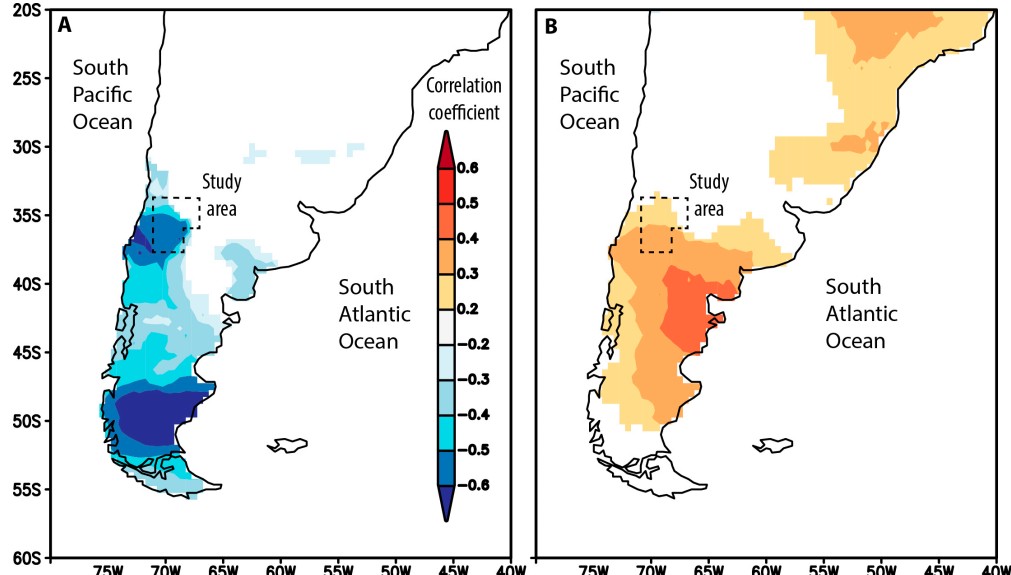

**Figure 2.** (**A**) Spatial correlations between the Southern Annular Mode (SAM) and seasonal (November–January) temperatures from CRU version TS4.01 (1957–2006). (**B**) Spatial correlations between SAM and seasonal (November–January) precipitation from CRU version TS4.01 (1957–2006).

The SAM reconstruction shows a period of predominant positive anomalies from the beginning of the record, 950 years cal BP, until about 550 years cal BP (SAM positive phase). After that, the negative phase persistently prevailed until the ca. 150 years cal BP, reaching extreme negative departures around 470 years cal BP (Figure 3; SAM negative phase). The persistence of the SAM positive phase implies the occurrence of warmer late spring–early summer temperatures in southern Mendoza during the first centuries of the past millennium. On the contrary, more negative SAM values prevailed between 550 and 450 years cal BP and persisted continuously, although less intense since ca. 150 years cal BP. This interval is associated with dominant below-average temperatures across the region. We assumed that the first centuries of the last millennium were relatively warm and dry, since the SAM positive phase relates to below-average rainfalls. Conversely, from the early 550 to 100 years cal BP, the climate during the late spring–summer season was predominantly colder and humid. This wetter, cooler period, which began in the region towards 550 years cal BP and persisted until approximately 100 years cal BP, was concurrent with glacial advances documented for the Andes in southern Mendoza [111] and northern Patagonia [112,113]. This cold and relatively humid period in the region, synchronous with the Little Ice Age in the North Hemisphere, contrasts with the warmer conditions recorded during the first three centuries of the last millennium. This warmer interval has been identified as the

Medieval Warm Period in the Northern Hemisphere. Its occurrence in South America is still under discussion [114].

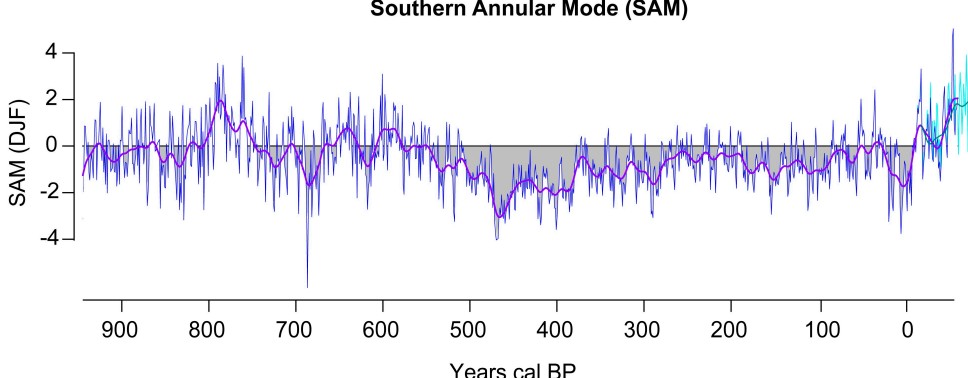

**Figure 3.** Reconstruction based on annual-resolution proxy records of the summer (DJF) Southern Annular Mode (SAM) or Antarctic Oscillation over the past 1000 years. The observed SAM record is shown in light blue. The reconstruction (thin blue line) is also shown in a smoothed version (violet line) using a digital filter (64-year spline). During the interval 950-550 years cal BP, associated with the Medieval Warm Period, the SAM oscillated around its long-term mean value. On the contrary, its values were mostly negative between 550 and 100 years cal BP (particularly 550-450 years cal BP), concurrent with the Little Ice Age.

### 3.2. Human Populations in Northwestern Patagonia: Trends in Radiocarbon SPD

The empirical SPD for the last millennium shows variation between 0.05 and 0.11 (Figure 4). We compared the empirical SPD with those from the exponential and uniform models and found that the empirical SPD is not different from model simulations, indicating a global p = 0.6 in both cases. There are no statistically significant empirical SPD deviations from the uniform and exponential null models. The null models take the noise of the sampling and calibration processes into account. This means that while there are no changes in the empirical SPDs through time; these changes could occur due to random noise caused by sampling and calibration. By the width of the grey bar (the null models), we consider that this is product of the small sample sizes. Based on these results, we assumed that this empirical SPD represents a relatively stable population through time.

The permutation test shows that Monte and Patagonia SPDs follow the critical envelopes of the null model (Figure 5). For the interval 700 to 300 years cal BP, we noted that increases in the empirical curve for the Monte were not related to variations in the Patagonia SPD. However, after 350 years cal BP, the Monte empirical SPD declines whereas the Patagonia SPD increases (Figure 5). This opposite pattern in SPD could indicate that these deserts did not have the same human demographic response during the last 1000 years cal BP, but still the difference is not enough to be statistically significant.

### 3.3. Stable Isotopes (C and N) in Human Bone Collagen

We explored the stable isotope results obtained from 43 human bone samples—bone collagen and bone structural carbonate (Figure 6) (see Materials and Methods; Table S2). The northwestern Patagonia human bones have a $\delta^{13}C_{coll}$ mean of −16.0‰ (SD = 2.5; max. = −10.0‰, min. = −19.0‰). The $\delta^{13}C_{coll}$ range indicates a notable variation (9‰), direct or indirect, in $C_3/C_4$ plant consumption. For northwestern Patagonia, Gil et al. [80] proposed as the extreme to calculate diet on herbivorous bone collagen the following values: $\delta^{13}C$ −21.4% (100% $C_3$ in diet) and −7.5 (100% $C_4$ in diet). Based on these values, human bone $\delta^{13}C_{coll}$ from the northwestern Patagonia mean diet composition (direct or indirect) of $C_4$ plants is between 86% and 18% (mean 40%). The $\delta^{15}N$ mean is 10.3 ± 1.4‰ and ranges between 7.2‰ to 13.2‰ (Figure 7; Table 1 and Table S2), showing a significant variation. Based on the mean of 5.2 ± 1.2 ‰ for northwestern Patagonia guanaco, which could be considered as the

herbivorous regional trophic base line [80], the mean difference of 5.1‰ suggests a human diet focused on meat and not in vegetables. The regional human bone $\delta^{13}C_{carb}$ mean value is −11.0% ± 2.9‰, ranging between −5.1‰ and −14.9‰ (Figure 7; Table 1). It indicates a difference of 5‰ with $\delta^{13}C_{coll}$; the correlation between $\delta^{13}C_{carb}$ and $\delta^{13}C_{coll}$ is 0.9.

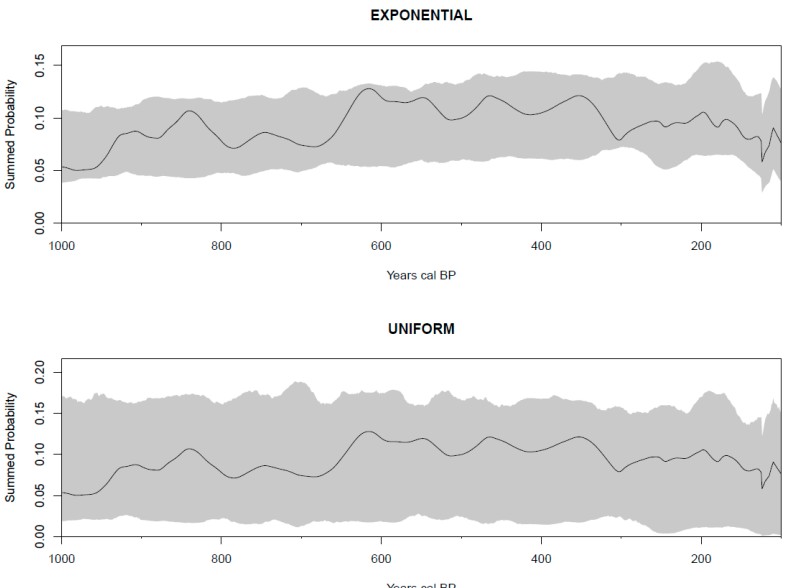

**Figure 4.** Summed probability distributions (SPDs) for northwestern Patagonia (southern Mendoza) and their fitted hypothetical null models (exponential and uniform). The black lines represent the empirical SPD while the grey areas are the 95% confidence interval for the simulated null models.

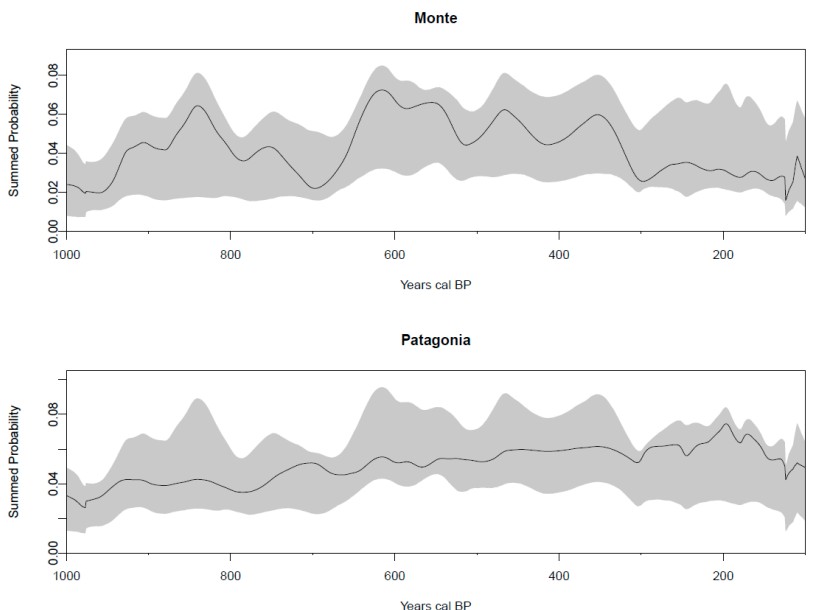

**Figure 5.** Radiocarbon SPDs for the Patagonia and Monte deserts over the past millennium.

Based on the chronological definition of the SAM positive (SAM+; 950 to 550 years cal BP) and negative (SAM-; 550 to ca. 150 years cal BP) phases, we used a GIS interpolation method to estimate differences in stable isotope on human bones over northwestern Patagonia. Nine human bones from Monte Desert and nine from Patagonia Desert composed the SAM positive phase samples. Human bones assigned to the SAM negative phase included nine and sixteen samples from the Monte and Patagonia deserts, respectively. Table 1 shows the basic statistics for $\delta^{13}C_{coll}$, $\delta^{13}C_{carb}$, and $\delta^{15}N$

from human bones according to the desert environments and SAM phases. Figure 6 displays the relationships between $\delta^{13}C_{coll}$ and $\delta^{15}N$ according to the deserts and SAM phases. In the Monte, two groups are recognized. The first Monte group shows $\delta^{13}C_{coll}$ between −10‰ and −15‰ and $\delta^{15}N$ between 10‰ and 9‰. The second Monte group shows $\delta^{13}C_{coll}$ between −15.3‰ and −19% and higher $\delta^{15}N$ between 10‰ and 13‰. A third group composed of human bones from the Patagonia Desert includes mostly (ca. 90% of the samples) individuals with lower $\delta^{13}C_{coll}$ values between −15.5‰ and −19.5‰, and highly variable $\delta^{15}N$ between 8‰ to 13‰. Inferred from these results, the first group corresponds to individuals with higher consumption of $C_4$ in their diets, possibly maize, than the other two groups. The last two groups have a minor, if any proportion of $C_4$. During the SAM positive phase, the human bone samples were mostly (66%) conformed by the first Monte group. In contrast, during the SAM negative phase, the human bones include all individuals from the third group and most of the second group (Figure 6).

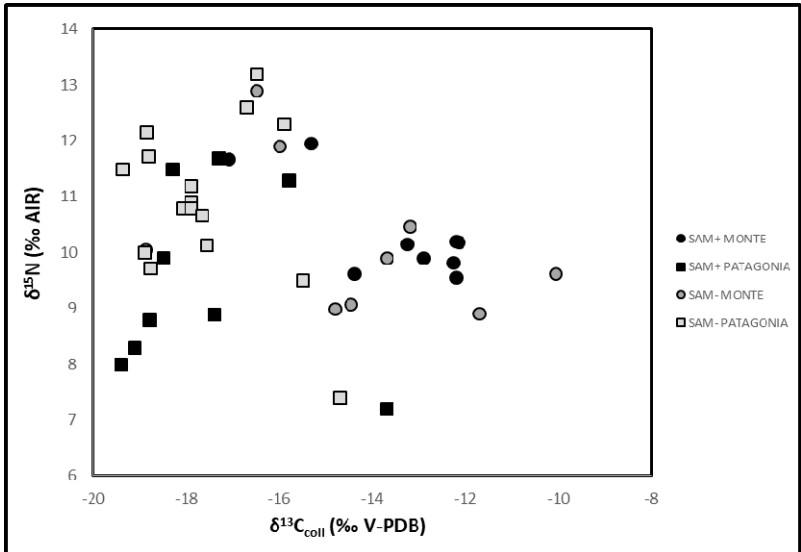

**Figure 6.** Individual variations in human bone collagen $\delta^{13}C_{coll}/\delta^{15}N$ according to desert (Monte/Patagonia) and SAM phase (positive/negative).

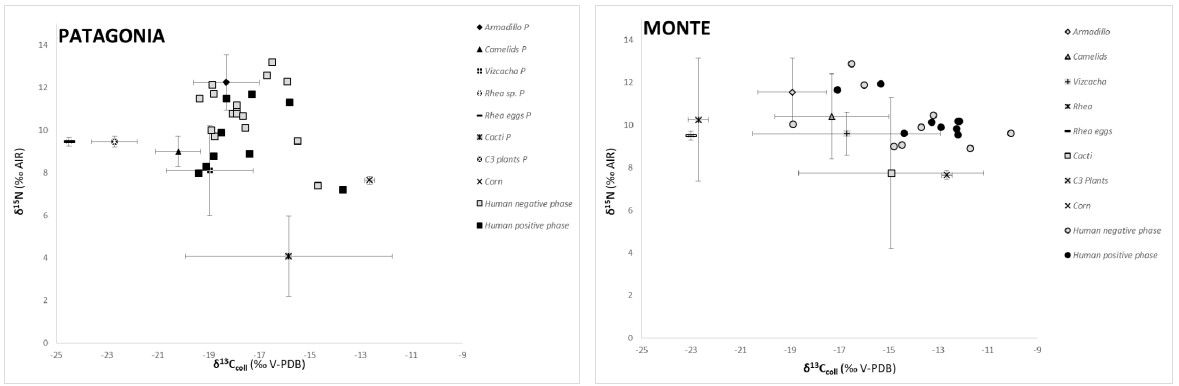

**Figure 7.** Regional resource trends in $\delta^{13}C$ and $\delta^{15}N$ for the Monte and Patagonia deserts (data base from Gil et al. [80]) and their relationships with human bone collagen isotopes (Table S2).

Figure 7 presents the relationship between human bone collagen stable isotopes with regional plants and animals stable isotopes (details in Gil et al. [97]). We grouped all data according to desert type and SAM phase. Individuals from the Patagonia Desert have higher $\delta^{15}N$ values than plants, indicating a diet largely based on meat. In addition, most individuals (with the exception of two) in the Patagonia Desert show lower $\delta^{13}C$ than $C_4$ plants and cacti (Figure 7). We observe two individual

groups in the Monte Desert samples: one closely related to guanaco/vizcacha and armadillo resources, and another to maize/cacti. This suggests different diets between the groups. A higher consumption of cacti and maize by the second group could be associated with higher human $\delta^{13}C$ values. However, the presence of cacti is not significant in the archaeobotanical record nor in the assessments of the isotopic mixing model diet [97]. We suggest that maize was the primarily resource responsible for enriching these human diets.

**Table 1.** Basic statistic in $\delta^{13}C_{coll}$, $\delta^{13}C_{carb}$, and $\delta^{15}N$ from human bones in southern Mendoza. Results are arranged by deserts (Monte and Patagonia) and SAM phases (positive and negative; see details in main text).

| Descriptive Statistics | SAM Positive Phase | | | | | | SAM Negative Phase | | | | | |
|---|---|---|---|---|---|---|---|---|---|---|---|---|
| | MONTE | | | PATAGONIA | | | MONTE | | | PATAGONIA | | |
| | $\delta^{13}C_{coll}$ | $\delta^{13}C_{carb}$ | $\delta^{15}N$ | $\delta^{13}C_{col}$ | $\delta^{13}C_{carb}$ | $\delta^{15}N$ | $\delta^{13}C_{col}$ | $\delta^{13}C_{carb}$ | $\delta^{15}N$ | $\delta^{13}C_{col}$ | $\delta^{13}C_{carb}$ | $\delta^{15}N$ |
| N | 9 | 9 | 9 | 9 | 9 | 9 | 9 | 9 | 9 | 16 | 16 | 16 |
| Min | −7.1 | −12.3 | 9.6 | −19.4 | −14.5 | 7.2 | −8.9 | −13.5 | 8.9 | 9.4 | −14.9 | 7.4 |
| Max | −12.1 | −6.5 | 12.0 | −13.7 | −7.6 | 11.7 | −10.1 | −5.1 | 12.9 | −14.7 | −8.2 | 13.2 |
| Mean | −13.5 | −8.4 | 10.3 | −17.6 | −12.9 | 9.5 | −14.4 | −8.8 | 10.2 | −17.6 | −12.8 | 10.9 |
| SD | 1.7 | 1.8 | 0.9 | 1.8 | 2.1 | 1.7 | 2.6 | 2.6 | 1.4 | 1.4 | 1.9 | 1.4 |
| Median | −12.9 | −8.0 | 10.2 | −18.3 | −13.6 | 8.9 | −14.5 | −7.9 | 9.9 | −17.9 | −13.1 | 10.9 |

Figure 8 compares the modelled continuous spatial distribution of $\delta^{13}C_{carb}$ and $\delta^{15}N$ contrasting the SAM positive and negative phases. The interpolation of $\delta^{13}C_{carb}$ indicates a mean highest value during the SAM positive phase, with most of the area between −7‰ and −8.5‰. In the SAM negative phase, the individuals show values predominantly between −12‰ and −14‰. In addition, the $\delta^{15}N$ also show contrasting spatial patterns. Lowest $\delta^{15}N$ values, between 7‰ and 8‰, occurred during the SAM positive phase. On the contrary, highest $\delta^{15}N$ values, between 11‰ and 13‰, were recorded during the SAM negative phase and cover most of the area. Combining both stable isotopes patterns, we infer a more plant-focused diet with stronger focus on $C_4$ resources during the SAM positive than the negative phase.

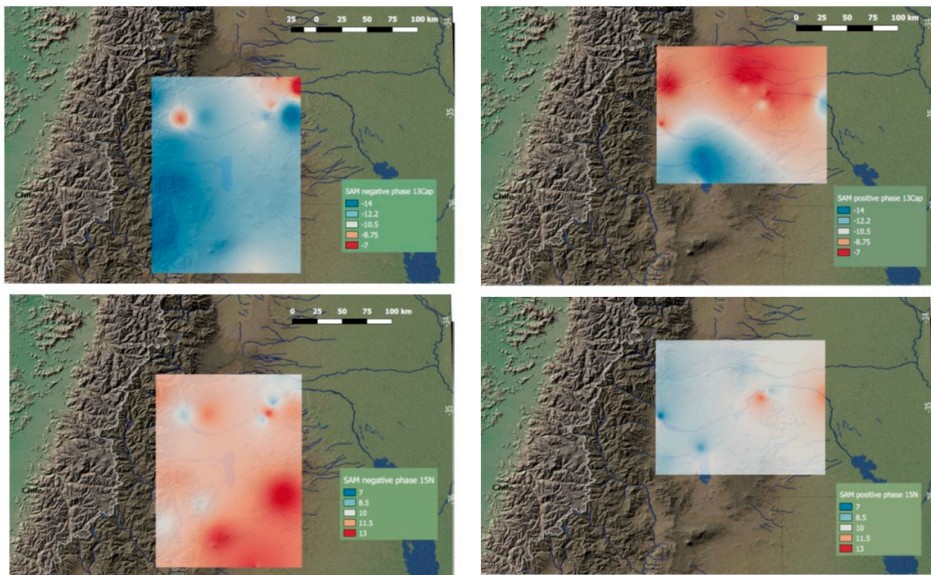

**Figure 8.** Northwestern Patagonia–southern Mendoza isoscapes (IDW interpolation) segregated by SAM phase (see details in the text).

*3.4. Artiodactyl Index*

The Artiodactyl Index (AI) shows no significant changes through time. The relationship between time (years cal BP) and AI is not significant (r = −0.2; p = 0.3). However, the regional pattern shows

that guanaco dominated faunal assemblages after 550 years cal BP, concurrent with a dominant SAM negative phase (AI > 0.9). In contrast, AI ranges between 0.05 and 0.9 during the previous SAM positive phase (Figure 9). The Monte Desert shows lower AI values in comparison with the Patagonia Desert. It is important to highlight that all samples dating to SAM negative phase are from the Patagonia Desert. SAM positive phase zooarchaeological samples from the Monte Desert are taxonomically richer and more diverse than the Patagonia Desert samples [45]. The guanaco-dominant diet in the Patagonia Desert during the SAM negative phase reflects access to the highest-ranked resources. The observed trend in the Patagonia Desert zooarchaeological assemblages explains the highest values of $\delta^{15}$N in human samples from the same period noted in Figures 6 and 9. At the same time, the highest variation and lowest AI during the SAM positive phase are concordant with lower $\delta^{15}$N and enriched $\delta^{13}$C, indicating a more variable and broader diet, which included a greater focus on $C_4$ resources, such as maize, during the SAM positive phase.

## 4. Discussion

After 2300 years cal BP, the human population size in northwestern Patagonia changed its low growth previous trajectory and reached levels never seen before [15]. Cultigens were not used homogenously in northwestern Patagonia. In addition, probably their use also shifted across the following 2300 years cal BP [9]. Maize productivity is highly variable as a consequence of inter- and intra-annual variation in temperature and precipitation [115–119]. Variation in productivity affects its energetic value and order in the resources rank. Extremes temperatures and moisture have negative effects. Low temperatures during the seedling-emergence period cause crop damage by late and early frosts. Heat stress during the critical flowering period can reduce or eliminate yields. Demerit et al. [117] and Benson et al. [115] proposed the use of the corn heat unit or growing degree days (GDD) to explore this temperature constriction. Their scheme assumes that a plant matures by a daily accumulation of heat units. From this perspective, changes in temperature can affect the geographic limit and the productivity of corn by reducing the number of GDD's accumulated per day and the length of the growing season. Either or both processes can lead to reduced yields and even total crop failure.

During SAM positive phase (from ca. 950 to ca. 550 years cal BP), lower summer precipitation but higher temperatures prevailed in northwestern Patagonia, likely increasing the GGDs. Isotopic markers of human diet indicate temporal variation in stable carbon and nitrogen isotopes (Figure 9D–F). Fitted Loess curves show a $\delta^{13}C_{coll}$ increase, ranging between −12‰ and −13‰ over the SAM positive phase (Figure 9E). The higher $\delta^{13}C_{coll}$ responds to an increase in the consumption of $C_4$ resources, including maize (Figure 7). Both deserts, Monte and Patagonia, have similar trends but their means are different. Monte shows higher $\delta^{13}$C than Patagonia, which in turn presents more stable values than Monte (Figure 9D,F). Similarly, in both deserts, bone collagen $\delta^{15}$N averages in ca. 10‰ (Figure 9D). Since the SAM positive phase is associated with higher summer temperatures and lower precipitation, higher $\delta^{15}$N values should be expected in resources to this SAM positive phase [120–122]. Recent analyses from guanaco bone collagen show increased $\delta^{15}$N between 1200 and 600 years cal BP in the Monte Desert, but not in the Patagonia Desert. The increased $\delta^{15}$N was associated with more extreme aridity conditions in the Monte [123]. If this is the case, the herbivorous $\delta^{15}$N base line increased from ca. 5‰ (Figure 7) to 9‰ in the Monte Desert, but not in the Patagonia Desert (Figure 4, [123]). We assume that human diet in the Monte Desert, between 900 and 600 years cal BP, had a larger component of plants than in the Patagonia Desert, despite people occupying both deserts had similar $\delta^{15}$N values.

The long-term SAM negative phase, between 550 years cal BP to 100 years cal BP, is associated with increased summer precipitation and lower summer temperatures. After 550 years cal BP, the human isotopic diet changed both in $\delta^{13}$C and $\delta^{15}$N (Figure 9D–F). In the Monte Desert there is a drop in $\delta^{13}C_{coll}$, from −13‰ to −17‰. A similar trend in $\delta^{13}C_{carb}$ is associated with increase in $\delta^{15}$N, from ca. 9.5‰ to ca. 12‰. The decrease in $\delta^{13}$C suggests a drop in $C_4$ consumption; the increase in $\delta^{15}$N supports a boost in meat consumption. In the Patagonia Desert, the $\delta^{13}C_{coll}$ is stable and similar to

those recorded during the SAM positive phase, from −13‰ to −17‰, with a similar trend in $\delta^{13}C_{carb}$. In contrast, there is an increase in $\delta^{15}N$, from ca. 9‰ to ca. 11.5‰. According to De Fina et al. [124], maize in Mendoza shows a decrease in the production of mature grains when the mean temperature of the warmest month is lower than 19.5 °C, and maize production fails completely when summers shorten. We assume that during the SAM negative phase, summers were shorter and impacted negatively on maize production. At the same time, wetter and cooler summers could have increased primary productivity. This implies an increase of wild plants and therefore in animal biomass.

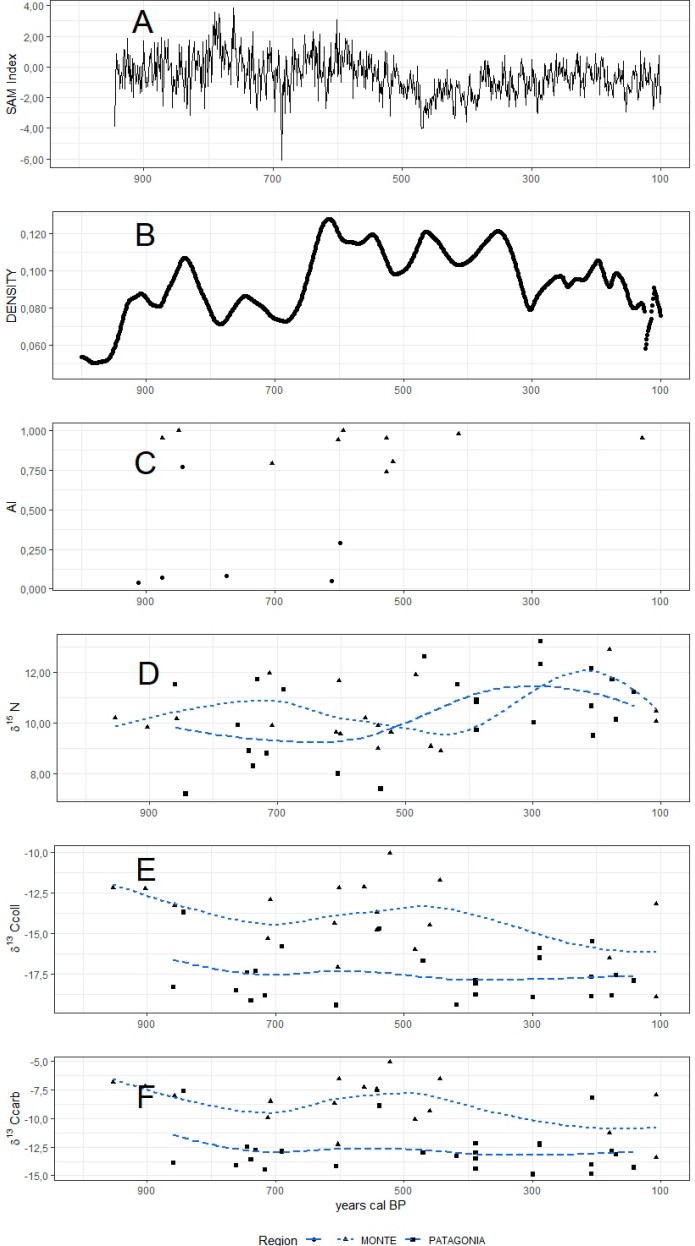

**Figure 9.** Multiproxy temporal analysis over the past millennium. (**A**) SAM variations from Figure 3; (**B**) radiocarbon SPD based on Figure 4; (**C**) Temporal variations for the Artiodactyl Index (AI); (**D**) long-term $\delta^{15}N$ variations using the LOESS filter (individual data segregated by desert); (**E**) $\delta^{13}C_{coll}$ temporal trend using the LOESS filter (individual data segregated by desert); (**F**) $\delta^{13}C_{carb}$ temporal trend using the LOESS filter (individual data segregated by desert).

The cooler and wetter summer could have induced a wild fauna drop in $\delta^{15}$N due to changes in animal physiological responses to climate [120]. If so, the recorded changes in human bone collagen $\delta^{15}$N is contrary to our expectation based on climate. It is a change in the human diet as far as $\delta^{15}$N increased instead to decline as expected by the SAM negative phase. In such case, this increase in $\delta^{15}$N indicates a diet more focus on meat than in plants.

Variations in agricultural strategies are influenced by the effects of local environmental conditions on marginal gains, for time spent in maize production, and the abundance of high-ranked wild resources [57]. They are not influenced by fluctuations in harvest yields. We expect more agricultural investment with decreasing opportunities to exploit highest-ranked items, such as guanaco in our region [43]. We expect less time spent on farming if there is an increase in alternative economic opportunities [125]. The decline in $\delta^{13}$C to −18‰ indicates a reduction in maize consumption, the more ubiquitous domesticated plant in the region. This drop in maize could mean less engagement in farming.

Climatic changes associated with the SAM phases could have triggered the recorded shifts in diets. Cool summers in northwestern Patagonia during the SAM negative phase represent a major risk for farming, particularly based on maize production [119]. However, the increase in summer precipitations under the same negative phase could have facilitated the increase in primary and secondary productivity. Other things being equal, if maize farming was more expensive, and at the same time wild resources increased, we can expect a change in the diet. This change implies a retraction of the farming frontier to the north and an internal change in the population dynamics.

The regional population dynamic did not change after the drop in maize consumption; on the contrary, it remained stable throughout the last millennium. Low-level maize production was probably implemented as a supplement to a diet based on wild resources that sustained the highest population levels observed in the last millennium. At the same time, regular to low consumption of maize indicates a highly variable human diet. Furthermore, we sustain that the decline in maize production was likely associated with an increase in wild resource productivity. If this were not the case, and the wild resources did not increase, a drop in population should have been recorded, a scenario that did not happen.

Figure 4 shows a drop in the empirical northwestern Patagonia SPD after 300 years cal BP. However, this drop is not significant when compared with the uniform and empirical null models. Spaniards arrived in Mendoza in 1551, and Mendoza city was founded in 1561 [126]. Peripheral areas, such as southern Mendoza, were occupied effectively by Europeans at the beginning of the 19th century. A change in human subsistence has been interpreted as a response to a declined population caused by European diseases and pathogens [107,127–129]. This post 300 years cal BP drop could be significant based on historic information. If this was the case, the SAM negative phase plus a drop in human demography could impulse the same pattern: a return to a diet based on high-ranked wild resources.

Our results confirm that the northwestern Patagonia farming border was dynamic through time and space. Interactions between climate variations, ecosystem productivity, and human organization could be invoked to explain past changes in the diet of the population that occupied these territories. The increase in primary productivity triggered during the SAM negative phase could impel a significant reorganization of the human/environment system after 550 years cal BP, in which human population size was maintained despite changes in subsistence strategies and diet. The scenario post-550 years cal BP shows a shift from a system that relied more on domestic plants, and perhaps to some degree on agriculture, towards a system with emphasis on hunting and gathering, as noted during historical times by chroniclers. In an area where the environmental conditions make farming a high-cost activity—a "low rank, low yield" in terms of Winterhalder and Kennett [57]—any change in the system can impact human decision-making. Therefore, the balance between resources can shift from a system more concentrated on low-level food production to another focused on hunter-gathering and wild resources exploitation. This would imply significant flexibility in adaptation strategies that are extremely sensitive to climate changes.

**Supplementary Materials:** The following are available online at http://www.mdpi.com/2571-550X/3/2/17/s1, Table S1: Radiocarbon database Northwestern Patagonia (South Mendoza), Table S2: Human samples information from northwestern Patagonia with stable isotope values, references, contextual and chronological information; Table S3: Artiodactyla Index northwestern Patagonia.

**Author Contributions:** Conceptualization, A.F.G., R.V., C.O., F.R.F., C.C.A., E.A.P., and G.N.; methodology, R.V., G.N. and A.F.G.; software, R.V., G.N. and A.F.G.; validation, F.R.F., C.C.A., and C.O.; formal analysis, A.F.G., R.V., and C.C.A.; investigation, A.F.G., R.V., C.O., F.R.F. and E.A.P.; data curation, E.A.P. and C.C.A.; writing—original draft preparation, A.F.G., R.V., and F.R.F.; writing—review and editing, C.C.A., C.O., E.A.P. and G.N.; visualization, C.C.A. and E.A.P.; supervision, A.F.G.; project administration, A.F.G.; funding acquisition, A.F.G. and R.V. All authors have read and agreed to the published version of the manuscript.

**Funding:** This research was funded by ANPCyT (grant PICT-2016/2667) and CONICET (grant PIP-11220150100342).

**Acknowledgments:** This research was funded by CONICET and ANPCYT. This research is part of the Past Global Changes (PAGES) PalEOclimate and the Peopling of the Earth (PEOPLE3K) network. We thank to LIECA lab (Ings. Gisela Quiroga and Armando Dauverné) for stables isotopes measurement and to Andrew Ugan for English review and comments about ideas and data presented in this paper. We acknowledge the use of imagery provided by services from NASA's Global Imagery Browse Services (GIBS), part of NASA's Earth Observing System Data and Information System (EOSDIS). Eugenia Ferrero (IANIGLA-CONICET) helped with preparation of Figure 3. This paper born as part of the PEOPLE3K workshop in Los Reyunos (May 2018), we thank to the participant by stimulate debates and the data exploration. A preliminary version of this paper was presented in INQUA Meeting, Dublin 2019. Thank to Erick Robinson and Jacob Freeman for exchange and comments about this topic. We thank to Museo de Historia Natural de San Rafael and Universidad Tecnológica Nacional Regional San Rafael. To editor of this especial volume, Encarni Montoya and Bronwen S. Whitney for invitation. We strongly highlight the contribution of four reviewers for their comments and criticism that helped to improve this paper. R.V. and AG were partially supported by BNP Paribas through the THEMES project.

**Conflicts of Interest:** The authors declare no conflict of interest.

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
