# Peer review of "Between Foragers and Farmers: Climate Change and Human Strategies in Northwestern Patagonia"

_quaternary, doi:10.3390/quat3020017_

Round 1

Reviewer 1 Report

I found the paper to be overall well written and well described. Introduction is informative and the aims of the study are clearly stated. Methods used in the study are relevant.

I listed below some general comments. I made specific comments/suggestions/corrections in the pdf file.

General comments:

Using two ways of expressing age (years BP and cal. years BP) can create confusion for readers.

Please use the same expression for age in the Figures as in the text (cal. years BP not Years cal BP). Also use dot after cal.

Please use Past Tense when describe events from the past.

Author Response

POINT 1: I found the paper to be overall well written and well described. Introduction is informative and the aims of the study are clearly stated. Methods used in the study are relevant. I listed below some general comments. I made specific comments/suggestions/corrections in the pdf file.

Response 1: We read your comments in the PDF and we introduced most of the changes you suggested to us.

POINT 2: Using two ways of expressing age (years BP and cal. years BP) can create confusion for readers.

Response 2: We improve this mistake and we change all age expressions to years cal BP.

POINT 3: Please use the same expression for age in the Figures as in the text (cal. years BP not Years cal BP). Also use dot after cal.

Response 3: We corrected the Figures and were consistent in the use of the same time scale and denomination used in the text: years cal BP.  We do not use dot after “cal” because the most popular use not use it (see Rcarbon, just an example use it without dot). But just in case we are open to change any way.

POINT 4: Please use Past Tense when describe events from the past.

Response 4: We reviewed the English grammar.

Reviewer 2 Report

Review Gil et al. Hunter-gatherer border in NW Patagonia
This paper discusses the potential correlations and causations between changes in human demography, diet, climate and environment over the past 2000 years in Patagonia. To this purpose, it adopts a multi-proxy approach, which is a sound methodological choice. However, the analysis of the individual categories of evidence is problematic and does not support the suggested interpretation. I recommend this paper to be rejected.
Intriguingly, whilst the paper sets up nicely the archaeological situation, the research questions, briefly mentioned at the end of the first paragraph, are not very precise, and not set within any clear theoretical framework. This leads to some imprecision, which is damageable for the overall argument. For instance, the authors mention a lot the question of spread of farming and border between farming and foraging communities as these were interchangeable issues, which clearly are not.
One of the key variables explored here is human demography, approached through the use of summed probability distributions of 14C dates. Arguably, the dataset is not very big, especially when considered at a regional basis. Although, technically, the used R package rcarbon can overcome this sample zise issue, this should not come at the expense of a basic description of the possible research biases / main categories of sites dated in the area, something which cannot be found in the present text. More worryingly, in this particular instance, the small sample size does limit the scope of the analysis, as reflected by the size of the MCMC envelopes, so that, in all instances, the analysis does not actually provide any statistically robust signal (as pointed out by the very high p-values presented in table 1), a point missed by the authors It is alo noticeable that the authors have decided to apply a range of models to the SPDs, which in itself is a good thing, but without questioning the value or underlying reasons for selecting these particular models: testing a SPD against an exponential or a logistic null does not have the same implications, and the corresponding discussion is absent of the text. Anyway, given the high p-values, contrary to the authors seem to think, the only reasonable inference to be made here is that the SPDs do not present any pattern at all, thus undermining one of the key assumptions on which the paper is based. Likewise, using a permutation test on two groups is of limited interest (as they will behave symmetrically). It is noteworthy that this procedure does not work using a null model, but rather an average, another key methodological point misunderstood by the authors. It must also pointed out that the corresponding rcarbon function privides a p-value, not reported here. Anyway, the only result - questionable as per above - is that there is a 50 years period where both regions differ, which is then abusively interpreted on line on line 399 as a statement that both regions experience different demographic responses during the last 1000 years. Lastly, figure 4 plays a big role in the overall argument, but comes without background discussion of the local research tradition, which could equally impact upon the SPDs. It ought to be simply replaced by a presentation of the local archaeology, or becomes the focus of an entire, complete analysis in itself. For instance, why not testing this SPDs against any null model?
The section on stable isotopes reads as a series of technical paragraphs on extracting collagen and measuring a signal, but contains only limited, if any, information regarding the nature of the signals to be tested. This section does need entire re-writing. One would want to know information about uncertainty, site distributions over time and space (beyond broad categorisations in main regions), nature of the sites sampled.
The stable isotopes data are partly analysed, as per fig.7 using the LOESS technique. I would like the authors to mention precisely the settings which they have used to perform this analysis (ie size of span), and also be explicit about the nature of the envelopes (I think it is a 95% confience interval, though admittedly the corresponding ggplot2 documentation is a bit sparse). I must also highlight the fact that the description of the technique is a mere copy-paste (I did not say plagiarism but...) from the corresponding wikipedia entry... When it comes to the interpretation of fig. 7, the authors entirely focus on the mean signal, and fail totally to consider the envelope. They also fail to note and discuss the abundance of outliers in all cases, which is very unfortunate given how sensitive the technique is to these. At the end, their reading is no more than eye-balling a graph to spot wide trends, for instance focusing on the 5550BP date inferred by the SAM signal, but which does not register in any way in the SPD. More traditionally, the authors also use t-tests (but did you check for the normality of each sub-sample?) to compare for C and N related signals, though this is not done in any systematic way. I must also highlight that, on lines 521 and following, the authors briefly discuss the possibiity that the change in N signal might be generalised (rather than specific to humans), but this is not done systematically / statistically.
Lastly, the AI index seems an interesting technique, though its descriotion is unclear (ie sum of what? NISP, %?). The description offered on lines 282-3 repeats lines 278-9, pointing to a lack of text-editing. Once more, the text is limited when it comes to describing the underlying dataset and its limits. Fig 7E shows region-based differences, but suggests nothing through time: this could be easily tested in a statistical way.

Author Response

POINT 1: This paper discusses the potential correlations and causations between changes in human demography, diet, climate and environment over the past 2000 years in Patagonia. To this purpose, it adopts a multi-proxy approach, which is a sound methodological choice. However, the analysis of the individual categories of evidence is problematic and does not support the suggested interpretation. I recommend this paper to be rejected.Intriguingly, whilst the paper sets up nicely the archaeological situation, the research questions, briefly mentioned at the end of the first paragraph, are not very precise, and not set within any clear theoretical framework. This leads to some imprecision, which is damageable for the overall argument. For instance, the authors mention a lot the question of spread of farming and border between farming and foraging communities as these were interchangeable issues, which clearly are not.

Response 1: Certainly our first version had a confuse focus. We modify it, and made emphasis on the stability of farming/hunter-gatherers border in northwestern Patagonia.  At the same time, we expanded our text, included our research question in a theoretical perspective, and generated expectations about the farming/hunter-gatherer border.

POINT 2: One of the key variables explored here is human demography, approached through the use of summed probability distributions of 14C dates. Arguably, the dataset is not very big, especially when considered at a regional basis. Although, technically, the used R package rcarbon can overcome this sample size issue, this should not come at the expense of a basic description of the possible research biases / main categories of sites dated in the area, something which cannot be found in the present text. More worryingly, in this particular instance, the small sample size does limit the scope of the analysis, as reflected by the size of the MCMC envelopes, so that, in all instances, the analysis does not actually provide any statistically robust signal (as pointed out by the very high p-values presented in table 1), a point missed by the authors It is alo noticeable that the authors have decided to apply a range of models to the SPDs, which in itself is a good thing, but without questioning the value or underlying reasons for selecting these particular models: testing a SPD against an exponential or a logistic null does not have the same implications, and the corresponding discussion is absent of the text. Anyway, given the high p-values, contrary to the authors seem to think, the only reasonable inference to be made here is that the SPDs do not present any pattern at all, thus undermining one of the key assumptions on which the paper is based. Likewise, using a permutation test on two groups is of limited interest (as they will behave symmetrically). It is noteworthy that this procedure does not work using a null model, but rather an average, another key methodological point misunderstood by the authors. It must also pointed out that the corresponding rcarbon function privides a p-value, not reported here. Anyway, the only result - questionable as per above - is that there is a 50 years period where both regions differ, which is then abusively interpreted on line on line 399 as a statement that both regions experience different demographic responses during the last 1000 years. Lastly, figure 4 plays a big role in the overall argument, but comes without background discussion of the local research tradition, which could equally impact upon the SPDs. It ought to be simply replaced by a presentation of the local archaeology, or becomes the focus of an entire, complete analysis in itself. For instance, why not testing this SPDs against any null model?

Response 2: We accept most of the recommendations and observations made by the reviewer. First, we updated our radiocarbon database, and now we include information regarding the category of site dated (Table S1). We modify the methodology text and we added information about possible research biases. Our database is small any way but, as we mention in the text, considering the size of the area, and the period of time, it has a similar sample size (or bigger) that the minimum number recommended by Williams (2012). The SPD model that can be rejected or modified with the advance of the research in the area. At the moment is one line of evidence (not the only evidence) to support the paper general argument.

***In this new version we compare the las 1,000 years cal BP with two models as null hypothesis (Shennan et al 2013  and Timpson et al 2014), not just to know the way the demography changed, but also to identify periods of significant population change (Crema et al. 2016). As Crema et al. (2016) proposed, we choose the exponential and uniform models. We use the exponential distribution because represented both the temporally increasing taphonomic loss, and the long-term population increase observed in prehistoric populations (Crema et al. 2016; Shennan et al. 2013). The uniform model does not assume an exponential increase in the underlying population. It looks for significant deviations from a simpler “flat” model (Crema et al. 2016). This second null model is sensitive to evaluate rise-and-fall patterns.

As highlighted by the reviewer, in our older version (and in this version too) we do not observe any significant departure from the models. If so, as mentioned by the reviewer, we cannot inform any population change, and therefore reject any significant change. We consider that the empirical variation could be noise of the calibration curve. As an alternative hypothesis, we argument that the population during the last 1,000 years cal BP was relatively stable. Certainly, future increase in the radiocarbon data base will facilitate a new discussion of the regional trend.  

***The reviewer mentions that if we have not any population change our argument is not supported. We show in this new version that we do not need a population change to explain the possible causes of why the farming/hunter gatherer border changed through time. If we assume these groups as “low-level food production”, the incorporation or the rejection of cultivars, will be absorbed into the foraging economy with a minimal impact on hunting and gathering (Winterhalder and Kennett 2020).  

***We introduced in this version the p value for the SPD analysis, as recommended by the reviewer.

***As we change updated the radiocarbon data base. Our comparison between both deserts show non-significant change looking in this last 1,000 years cal BP. They show some difference in each empirical SPD but they are no statistically significant. 

***Figure 4 is updated in this new version. We included a deeper discussion in the text of Figure 4, which intends to illustrate the Late Pleistocene-Holocene regional population dynamic with a focus on the last 1,000 years cal BP. The intention is to look at the last 1,000 years in a broader temporal scale. We accepted the reviewer recommendation and compared the empirical curve with a uniform model as null hypothesis. It was a proper approach to detect significant deviations, which are discussed in the text.

POINT 3: The section on stable isotopes reads as a series of technical paragraphs on extracting collagen and measuring a signal, but contains only limited, if any, information regarding the nature of the signals to be tested. This section does need entire re-writing. One would want to know information about uncertainty, site distributions over time and space (beyond broad categorisations in main regions), nature of the sites sampled.

Response 3: We accepted the reviewer recommendation and included more details about the meaning of isotopic signals. We re-wrote the text, expanding the older version and including information about the framework to interpret the isotopic values in terms of the human diet in this region. In Table S2 there are references and contextual information about the human bone samples included in this analysis.

POINT 4: The stable isotopes data are partly analyzed, as per fig.7 using the LOESS technique. I would like the authors to mention precisely the settings which they have used to perform this analysis (ie size of span), and also be explicit about the nature of the envelopes (I think it is a 95% confidence interval, though admittedly the corresponding ggplot2 documentation is a bit sparse). I must also highlight the fact that the description of the technique is a mere copy-paste (I did not say plagiarism but...) from the corresponding wikipedia entry... When it comes to the interpretation of fig. 7, the authors entirely focus on the mean signal, and fail totally to consider the envelope. They also fail to note and discuss the abundance of outliers in all cases, which is very unfortunate given how sensitive the technique is to these. At the end, their reading is no more than eye-balling a graph to spot wide trends, for instance focusing on the 5550BP date inferred by the SAM signal, but which does not register in any way in the SPD. More traditionally, the authors also use t-tests (but did you check for the normality of each sub-sample?) to compare for C and N related signals, though this is not done in any systematic way. I must also highlight that, on lines 521 and following, the authors briefly discuss the possibility that the change in N signal might be generalized (rather than specific to humans), but this is not done systematically / statistically.

Response 4: We updated the regional human bone stable isotopes values. Accepting the reviewer comments, we change substantially this part of the manuscript. In this new version, we added a table with descriptive statistics, and a Figure that compare individuals, regions and SAM phases.  The previous Figure 7 was changed, updated and now is Figure 10.  We added a Figure 9 with a GIS analysis (IDW interpolation) that was not included in the older version. We tried to improve our English explanation in order to avoid text like “copy and paste” from other sources. Our discussion of Loess trends includes now a consideration about the outliers and variation, not only the “mean signal”, but our focus obviously remains in the “mean signal”.

POINT 5: Lastly, the AI index seems an interesting technique, though its descriotion is unclear (ie sum of what? NISP, %?). The description offered on lines 282-3 repeats lines 278-9, pointing to a lack of text-editing. Once more, the text is limited when it comes to describing the underlying dataset and its limits. Fig 7E shows region-based differences, but suggests nothing through time: this could be easily tested in a statistical way.

Response 5: In the new version we expanded this part of the paper.  Now, we explain more details, and we include references to read about the AI. We corrected the editing problem. We included a statistic test and we discussed the meaning of these trends in the new version.

Reviewer 3 Report

This is an important paper that brings about a whole suite of new set of data and provides an ample summary of the existing ones. The research methods are clearly and comprehensively detailed. Human population dynamics using the radiocarbon summed probability distributions is well argued and its relevance supported by earlier applications in this regard However, the  applicability of other methods for the studied phenomena have not been discussed in satisfactory detail making their heuristic value not comprehensively confirmed. The work is based upon  a solid number of radiocarbon dates making the temporal resolution of the studied phenomena justifiable in most cases. However, a chronological position of faunal assemblages is problematic when it is established on the basis of a single date only.

The paper uses the recent data published from the region to argue more work should be done on the response / resilience in the face of these changes. The main problem with the paper is that the goals of the research are too broad and are ambiguously presented. The research hypothesis should be more focused. In one place one the authors argue that the paper is to ‘reconstruct longstanding climatic and archaeological changes’ and in other they want to ‘explore how changes in human strategies are modulated differentially with the climate structure’. An overview of the work on the advancement of farming is too narrowly focused. The presentation of the studied area lacks clarity, particularly for the reader not familiar with the local geography. Figure 1 needs to be re-drawn accordingly.

Measuring human response to climate change is a large and difficult topic to answer in the scope of one paper, and is not broken down into specific, smaller objectives. The relationship between human societies and climate change, and given that this topic is large and complex, would make the paper much stronger if explicitly proposes which specific climate variables are discussed here and which of these are the most relevant to the sites in the discussed region.  The clarity of the paper and its objectives is further blurred by using largely imprecise phrases such as ‘the role of human and environmental interaction’ and ‘the climatic and environmental structure variation’.

While the paper successfully reconstructed longstanding climatic and archaeological changes in the period in question, it largely failed in explaining how human strategies are modulated differentially with the climate structure. The provided discussion is more a summary of the independently analyzed datasets than the critical evaluation of the achieved results. As such, the largely failed to directly address the advocated research hypothesis and the last paragraph does not suffice in this respect. Consequently, the relations between hunter-gatherers and farmers are not satisfactorily scrutinized. This section certainly needs to be expanded to make the research questions explicitly targeted and critically evaluated.

Author Response

POINT 1: This is an important paper that brings about a whole suite of new set of data and provides an ample summary of the existing ones. The research methods are clearly and comprehensively detailed. Human population dynamics using the radiocarbon summed probability distributions is well argued and its relevance supported by earlier applications in this regard. However, the applicability of other methods for the studied phenomena have not been discussed in satisfactory detail making their heuristic value not comprehensively confirmed. The work is based upon a solid number of radiocarbon dates making the temporal resolution of the studied phenomena justifiable in most cases. However, a chronological position of faunal assemblages is problematic when it is established on the basis of a single date only.

Response 1: We expanded the methodology on other proxies used in this document. The new version has better explanations and sets clear expectations regarding the change or stability of human strategies and how different proxies can show these responses. Certainly, our fauna database has a weak link between dates and the assemblages. However, we assume the fauna databse represents time averages for the assemblages with centennial deviation. It is not possible for the zooarchaeological assemblages to have the same time resolution as stable human bone isotope. Therefore, both proxies represent a different time resolution. In the database, the fauna assemblages have a multicentennial resolution, and the human bones have a multidecadal resolution. We cannot compare both vis a vis, but we can compare both as multicentennial trends. This is what we propose in this new version.

POINT 2: The paper uses the recent data published from the region to argue more work should be done on the response / resilience in the face of these changes. The main problem with the paper is that the goals of the research are too broad and are ambiguously presented. The research hypothesis should be more focused. In one place one the authors argue that the paper is to ‘reconstruct longstanding climatic and archaeological changes’ and in other they want to ‘explore how changes in human strategies are modulated differentially with the climate structure’. An overview of the work on the advancement of farming is too narrowly focused. The presentation of the studied area lacks clarity, particularly for the reader not familiar with the local geography. Figure 1 needs to be re-drawn accordingly.

Response 2: In this new version we changed the text and made a more focused paper. We included a new Figure 1.

POINT 3: Measuring human response to climate change is a large and difficult topic to answer in the scope of one paper, and is not broken down into specific, smaller objectives. The relationship between human societies and climate change, and given that this topic is large and complex, would make the paper much stronger if explicitly proposes which specific climate variables are discussed here and which of these are the most relevant to the sites in the discussed region.  The clarity of the paper and its objectives is further blurred by using largely imprecise phrases such as ‘the role of human and environmental interaction’ and ‘the climatic and environmental structure variation’.

Response 3: We agree with the reviewer that the human response to climate change is an interesting and complex topic to research. In the new version we tried to focus in two variables that changed during the last 1,000 years cal BP: summer temperature and summer precipitation. In the new version we tried to clarify our goals and made the paper more focused and clearer. 

POINT 4: While the paper successfully reconstructed longstanding climatic and archaeological changes in the period in question, it largely failed in explaining how human strategies are modulated differentially with the climate structure. The provided discussion is more a summary of the independently analyzed datasets than the critical evaluation of the achieved results. As such, the largely failed to directly address the advocated research hypothesis and the last paragraph does not suffice in this respect. Consequently, the relations between hunter-gatherers and farmers are not satisfactorily scrutinized. This section certainly needs to be expanded to make the research questions explicitly targeted and critically evaluated.

Response 4: The new version expands a theoretical perspective and explains in detail the expectations regarding human population, resources, and climate in northwestern Patagonia. We organized the discussion from this perspective, trying to give an answer to our question regarding the stability of farming/foragers border.

Reviewer 4 Report

This manuscript shows paleoclimate change data in Northwestern Patagonia crossreferenced with C N isotopes results some from previous literature.

The quality control parameters employed to construct the models should be described in material and methods section. 

Furthermore if the obtained isotopes data are processed together with the results from previous literature this issue should be specified in the text.

The writing needs to be consistently improved. The comprehension of  some paragraphs is difficult due to the poor quality text composition .

Author Response

POINT 1: This manuscript shows paleoclimate change data in Northwestern Patagonia crossreferenced with C N isotopes results some from previous literature.The quality control parameters employed to construct the models should be described in material and methods section. Furthermore if the obtained isotopes data are processed together with the results from previous literature this issue should be specified in the text. The writing needs to be consistently improved. The comprehension of some paragraphs is difficult due to the poor quality text composition .

Response 1: We accepted the reviewer recommendations and we added this information in this new version. The English grammar was reviewed in order to improve this version.

Round 2

Reviewer 2 Report

This revised version of the paper is significantly better than the original submission. The revised theoretical focus and methodological presentation offer a more suitable framework for the discussion of the authors' result.

At this stage, my comments only concern very minor typos / editing issues:

- lines 46-50: repetitive sentences, please delete one or merge.
- line 193: editing required
- line 423: typo in formula of the AI

Author Response

We accept the three typos/editing changes recommended by the reviewer. 

Reviewer 4 Report

The manuscript shows interesting hypothesis about the past environmental populational dynamics in northwestern Patagonia. In this new submitted version, the authors present many data but is still difficult to understand the use and relation between data produced by the authors and data from references. In addition, I see that many concepts are expressed several times throughout the text, this make the work quite heavy to read. Some paragraphs are just a very long list of specimens, may a table of references would be enough. On my opinion the authors should present a shorter version of the manuscript were just important data to support their hypothesis should appear. Furthermore, in Methods section there should be more order, may two subsections dividing analytical techniques and statistics methods. Substantially the effort of the authors should be focused on make the manuscript easier to read avoiding the addition of contents that are not particularly necessary that instead break the fluency of the work. Finally, there are too many self-citations from some of the authors, on my opinion not always fundamentals.

Cheers

Author Response

  1. In this new submitted version, the authors present many data but is still difficult to understand the use and relation between data produced by the authors and data from references.

We consider this suggestion throughout the new revision and delated the SPD Analysis for the whole Holocene and delete part of the ecological description of srudy area.

  1. In addition, I see that many concepts are expressed several times throughout the text, this make the work quite heavy to read.

We deleted several (n=23) sentences that could be considered redundant across the text

  1. Some paragraphs are just a very long list of specimens, may a table of references would be enough.

We rewrote the paragraphs.

  1. On my opinion the authors should present a shorter version of the manuscript were just important data to support their hypothesis should appear.

By addressing points 2 and 6 we also committed to improve this point.

  1. Furthermore, in Methods section there should be more order, may two subsections dividing analytical techniques and statistics methods.

We created sub-section Summed probability distribution to organize the information better. We deleted several sentences which would be considered redundant or far to detailed to report the results.

  1. Substantially the effort of the authors should be focused on make the manuscript easier to read avoiding the addition of contents that are not particularly necessary that instead break the fluency of the work.

We delated several paragraphs and sentences to address this point.

  1. Finally, there are too many self-citations from some of the authors, on my opinion not always fundamentals

We deleted some self-citations in the text. Most of “self-citation” are about where the DataBase included in Supplementary Table (TS1, TS2, TS3) are first time published.

Round 3

Reviewer 4 Report

The manuscript has improved. Some paragraphs are still difficult to read. The authors should try to make the manuscript more fluent.

Author Response

We made other English grammar review and we have other garmmar review from a native external. Through more than 300 editions we worked for a more fluent paper:

1) We made sentences clearer and shorter. 

2) We have erased several criptic and redundant sentences. 

3) We checked conceptual consistency and style. 

4) We checked verb and subject coherence within sentences and paragraphs.

 5) We flipped sentence organization.

We consider the paper has improve, is shorter and clearer. We thank the reviews that made this progress possible.